# EMBERSim: A Large-Scale Databank for Boosting Similarity Search in Malware Analysis

**Dragoș Georgian Corlătescu**
CrowdStrike
dragos.corlatescu@crowdstrike.com

**Alexandru Dinu**
CrowdStrike
alexandru.dinu@crowdstrike.com

**Mihaela Găman**
CrowdStrike
mihaela.gaman@crowdstrike.com

**Paul Sumedrea**
CrowdStrike
paul.sumedrea@crowdstrike.com

## Abstract

In recent years there has been a shift from heuristics-based malware detection towards machine learning, which proves to be more robust in the current heavily adversarial threat landscape. While we acknowledge machine learning to be better equipped to mine for patterns in the increasingly high amounts of similar-looking files, we also note a remarkable scarcity of the data available for similarity-targeted research. Moreover, we observe that the focus in the few related works falls on quantifying similarity in malware, often overlooking the clean data. This one-sided quantification is especially dangerous in the context of detection bypass. We propose to address the deficiencies in the space of similarity research on binary files, starting from EMBER — one of the largest malware classification data sets. We enhance EMBER with similarity information as well as malware class tags, to enable further research in the similarity space. Our contribution is threefold: (1) we publish EMBERSim, an augmented version of EMBER, that includes similarity-informed tags; (2) we enrich EMBERSim with automatically determined malware class tags using the open-source tool AVClass on VirusTotal data and (3) we describe and share the implementation for our class scoring technique and leaf similarity method.

## 1 Introduction

Malware is employed as a cyber weapon to attack both companies as well as standalone users, having severe effects such as unauthorized access, denial of service, data corruption, and identity theft [1]. From signature-based malware detection [2], to more sophisticated machine learning (ML) techniques [3, 4], antivirus (AV) became a necessary layer of defense in this cyber-war. With unprecedented access to open-source tools that facilitate replicating existing malware, the number of similar malicious samples is constantly on the rise [5]. This poses a problem both from the perspective of Threat Researchers having to analyze more data each year, as well as for Data Scientists training ML models for malware detection. Including near-duplicate samples in new iterations of malware detectors is dangerous as it can introduce biases towards certain types of attacks. Thus, we can argue that being able to perform similarity search at scale is of utmost importance for the Cybersecurity world [6]. Given the vast amounts of data, machine learning is better equipped to hunt for patterns in the context of similarity search [7, 8, 9, 10, 11], rather than relying on heuristics [12, 13] or signatures [5, 14] as it has been the case until recently.

37th Conference on Neural Information Processing Systems (NeurIPS 2023) Track on Datasets and Benchmarks.

In this work, we propose to address the problem of similarity search in Windows Portable Executable (PE) files. Our choice of platform is driven by the vast majority of computers worldwide that use the Windows operating system [15]. Moreover, our motivation for approaching binary code similarity (BCS) is the challenging nature of this problem due to factors such as the absence of source code and binary code varying with compiler setup, architecture, or obfuscation [8, 16, 17]. We identified a scarcity of data in the space of BCS in Cybersecurity [17] and decided to address this by generating a benchmark for binary code similarity. Our work uses EMBER [18] – a large data set of PE files intended for malware detection – and a combination of automatic tagging tools and machine learning techniques as described below. Our hope is to enable future research in this area, with a focus on considering real-world complexities when training and evaluating malware similarity detectors.

Our contribution is threefold:

1. We release EMBERSim, an enhanced version of the EMBER data set, that includes similarity information and can be used for further research in the space of BCS.
2. We augment EMBERSim with automatically determined malware class, family and behavior tags. Based on these tags, we propose a generally-applicable evaluation scheme to quantify similarity in a pool of malicious and clean samples.
3. As a premiere for the Cybersecurity domain, we repurpose a malware classifier based on an ensemble of gradient-boosted trees [19] and trained on the EMBER data set, to quantify pairwise similarity [20]. We publish the relevant resources to reproduce, on any other sample set, both this similarity method as well as our malware tag scoring technique based on tag co-occurrence. Code and data at: `CrowdStrike/embersim-databank`.

The remainder of this paper is organized as follows. In Section 2 we position our research in the related context for binary code and pairwise-similarity detection. Section 3 develops on the data that this work is based on, with a focus on EMBERSim – the enhanced EMBER, augmented with similarity-derived tags. Section 4 introduces the tree-based method used for similarity and the tag enrichment workflow. The experimental setup and evaluation are discussed in Section 5. Finally, we conclude the paper with Section 6, giving an outlook of this research.

## 2 Related Work

In this section, we put our work in perspective with a few related concepts and domains. We regard our own method with respect to the landscape of binary code similarity in general and with Cybersecurity in particular. We briefly discuss our choice of similarity method and its applicability to this domain. Finally, we inspect the literature exploiting EMBER [18] – the data set used as support – and note a few insights.

Binary code similarity (BCS) has, at its core, a comparison between two pieces of binary code [17]. Traditional approaches to BCS employ heuristics such as file hashing [12, 21, 22], graph matching [23] or alignment [24]. A sane default in the tool set of a Threat Analyst for similarity search is ssdeep [25], a form of fuzzy hashing mapping blocks of bytes in the input to a compressed textual representation, and using a custom string edit distance as the similarity function. Alternatively, machine learning is often chosen to tackle similarity in binary files, both with shallow [26], as well as with deep models [7]. Deep Learning, in particular, has been extensively used for BCS in the past years, from simpler neural networks [7, 8, 27, 28] to GNNs [6, 29, 30] and transformers [16, 31]. However, only a handful of the BCS ML-powered solutions [17] tackle the Cybersecurity domain [32, 33]. Out of these, most papers address similarity in functions [7, 27, 29] rather than whole programs [28].

With the present work, we propose to fill in some gaps in BCS for Cybersecurity, with a focus on whole programs. We differentiate ourselves from related research through our choice of method – repurposing an XGBoost [34] based malware classifier for PE files to leverage pairwise similarity based on leaf predictions (i.e. also known as proximities) [20]. The possibility of extracting pairwise similarities from tree-based ensembles is suggested in a few different works such as Rhodes et al. [20], Criminisi et al. [35], Cutler et al. [36], Qi et al. [37]. Similarity, in this case, is determined by counting the number of times when two given samples end up in the same leaf node, as part of prediction [36]. Intriguingly, despite its effectiveness in Medicine [38, 39] and Biology [37], we did not find any correspondence of the aforementioned method in the literature for BCS, especially applied in Cybersecurity.

Literature shows a scarcity of data for similarity detection in PE files [17]. Most of the past works are validated only against one of the benign/malicious classes [40], with less than 10K [41, 42, 40] samples in the test set [17]. Thus, we propose to conduct our research on EMBER [18], a large data set (1M samples) originally intended for malware detection in PE files. Other works using EMBER as support usually focus on malware detection [43, 44, 45, 46]. Similar to our second contribution, Joyce et al. [47] use AVClass [48] to label the malicious samples in EMBER with the corresponding malware family. However, the focus in this work is on categorizing the malware, whereas we have a different end goal (i.e. similarity search) and workflow involving the aforementioned tags. To the best of our knowledge, we are the first to release an augmented version of a large scale data set with rich metadata information for BCS.

## 3 Data

EMBER [18] is a large data set of binary files, originally intended for research in the space of malware detection. Most of the works that use EMBER as support continue the exploration following the initial purpose for which this data has been released – malware detection [43, 44, 45, 46] and classification by malware family [47]. In the present work, we enhance EMBER with similarity-derived information. Our goal is to enable further research on the subject of ML-powered binary code similarity search applied in Cybersecurity – a rather low resource area as discussed in Section 2. Through this section, we briefly describe the data used in our experiments, as well as the new metadata added to EMBER as part of our contribution.

### 3.1 EMBER Overview

In this research, we use the latest version of the EMBER data set (2018, feature version 2)[1], containing 1 million samples, out of which 800K are labeled, while the remaining 200K are unlabeled. The labeled samples are further split into a training subset of size 600K and a test subset of size 200K, both subsets being perfectly balanced with respect to the label (i.e. benign/malicious). Based on the chronological order of sample appearance, a significant portion (95%) of the samples were introduced in 2018, with the data set primarily comprising benign samples prior to that period. Additionally, it is worth noting that the division of the data set into train and test subsets follows a temporal split that clearly delimits the two subsets. Overall, the EMBER repository clearly describes the data, contains no personally identifiable information or offensive content, and provides a readily applicable and easily customizable method for the feature extraction required for training and experimenting with new models on samples known to all vendors using VirusTotal (VT)[2].

### 3.2 EMBERSim – Metadata Enhancements

**EMBER with AVClass v2 Tags.** In order to validate and enrich the EMBER metadata, we query VirusTotal, and obtain results from more than 90 AV vendors, particularly related to the detection result (malicious or benign) and, where applicable, the attributed detection name. This step is necessary in order to assess potential label shifts over time given that detections tend to fluctuate across vendors and the extent to which this can happen has not been quantified in the original EMBER data. Next, following a procedure similar to the one outlined in EMBER[3], we incorporate a run of the open source AVClass[4] system [48, 49] on the collected VirusTotal data. One key distinction is that we use AVClass v2 [48], which was not accessible during the original publication of EMBER. First of all, with the well-defined and comprehensive taxonomy in AVClass v2 [48] we aim to obtain standardized detection names from vendors. Secondly, we make use of the new features in AVClass2 to rank the generated tags (given vendor results) and to categorize them into FAM (family), BEH (behavior), CLASS (class), and FILE (file properties). To give an idea about interesting malware tags, we list the 10 most prevalent values below:

- **CLASS**: grayware, downloader, virus, backdoor, worm, ransomware, spyware, miner, clicker, infector.

---

[1]https://github.com/elastic/ember

[2]https://virustotal.com

[3]https://camlis.org/2019/talks/roth

[4]https://github.com/malicialab/avclass

- **FAM**: xtrat, zbot, installmonster, ramnit, fareit, sality, wapomi, emotet, vtflooder, ulise.
- **BEH**: inject, infostealer, browsermodify, ddos, filecrypt, filemodify, autorun, killproc, hostmodify, jswebinject.
- **FILE**: os:windows, packed, packed:themida, msil, bundle, proglang:delphi, small, installer:nsis, proglang:visualbasic, exploit

**Tag enrichment via co-occurrence.** Following the tag enrichment procedure outlined in Subsection 4.2.2, we augment the list of tags in EMBER with new tags which frequently appear together, as indicated by the tag co-occurrence matrix constructed using the AVClass v2 capabilities.

**Obtaining tag ranking for evaluation.** We leverage the rich tag information extracted from AVClass and augment it with tag co-occurrence data, to generate the metadata for EMBERSim. Table 1 shows how the presence of tags is correlated with the original label. This ground truth metadata can be utilized to evaluate the efficacy of our proposed similarity search method described in Subsection 4.1. The approach we explored for evaluation is described in Subsection 4.2.1.

Table 1: Previous vs. current AVClass tag presence with respect to label. Red cells indicate potential disagreement among label & tag presence (e.g. benign files with tags & malicious files without tags).

| prev | missing | | present | | **All** |
|---|---|---|---|---|---|
| curr | missing | present | missing | present | |
| label | | | | | |
| unlabelled | 95,359 | 8,208 | 1,958 | 94,475 | 200,000 |
| benign | 378,792 | 21,208 | 0 | 0 | 400,000 |
| malicious | 2,542 | 8,891 | 6,264 | 382,303 | 400,000 |
| **All** | 476,693 | 38,307 | 8,222 | 476,778 | 1,000,000 |

Table 1 highlights potential disagreements between the current and previous version of AVClass attributed tags. These tag changes can occur for benign files in limited instances (less than 5% in our case) given that greyware/adware is hard to quantify as either benign or malicious and vendors can have different views on a particular sample. Another limitation which can be observed here is that AVClass cannot attribute tags to approx. 1% of malicious samples. This is due to the fact that these samples did not have additional metadata available from VT to process at the time of writing as well as at the time of the original data set creation.

## 4 Method

### 4.1 Binary Code Similarity Using Leaf Predictions

As mentioned in Section 1, this work proposes viewing similarity through the lens of leaf predictions. We regard this as one of our main contributions, provided that, as far as studied literature shows, the aforementioned method based on leaf similarity was not applied before in BCS for Cybersecurity.

The similarity method outlined in this subsection is based on leaf predictions, which means that we depend on training a tree-based classifier. Although the method itself can be applied to any tree-based ensemble, our specific choice of algorithm is an XGBoost ensemble, due to its relevance in both industry in general [50] and in academic research [51]. Given our choice of data set and the domain addressed in this work, the aforementioned XGBoost model is trained to differentiate between malicious and benign PE files. The training setup and model performance do not make the object of this research and are only briefly discussed in Section 5. Relevant to our work is how we employ the binary XGBoost classifier to quantify the pairwise similarity between samples.

Given two samples, $s_1, s_2$, and a trained tree-based model, we compute the leaf predictions represented as two vectors – $x_1$ corresponding to $s_1$ and $x_2$ corresponding to $s_2$. Each of the two vectors contains the indices of the terminal nodes (leaves) for each tree in the ensemble when passing the respective sample through the model, with $T$ being the total number of trees in the ensemble. Intuitively, the vector of leaf predictions can be viewed as a hash code, where each value at position $i$ (i.e. $x_1^{(i)}$ and $x_2^{(i)}$ respectively) corresponds to a specific region in the input space partitioned by the $i$-th tree. Therefore, we consider two samples to be similar if they follow similar paths through the trees (denoted by the indicator function), hence obtaining similar leaf predictions. For the remainder

of this paper, we refer to this approach as *leaf similarity* which gives a score formally, computed as:

$$\text{LeafSimilarity}(x_1, x_2) = \frac{1}{T} \cdot \sum_{i=1}^{T} \mathbb{1}[x_1^{(i)} = x_2^{(i)}] \tag{1}$$

## 4.2 EMBERSim Metadata Augmentation Procedure

In Section 3, we have outlined a few insights regarding the metadata enrichment performed on EMBER. The diagram in Figure 1 provides a high-level overview of the augmentation and evaluation procedures followed. The main steps are detailed in the continuation of this subsection.

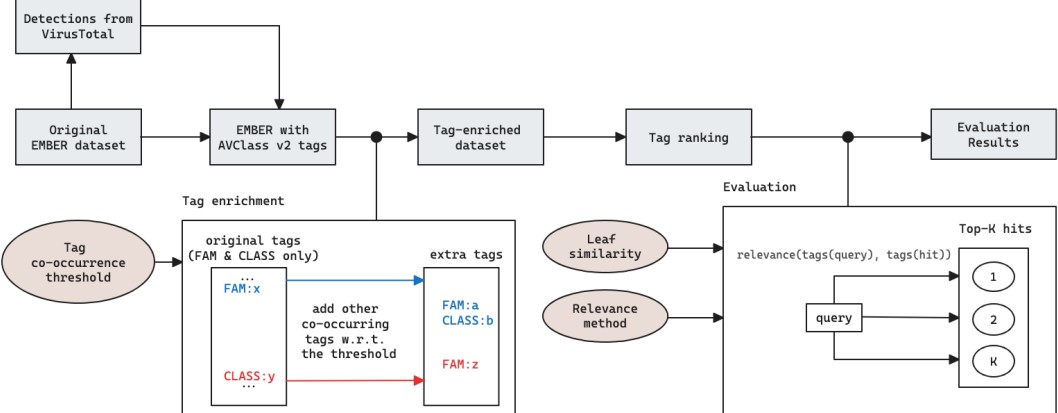

Figure 1: Summary of the data set enrichment and evaluation process.

### 4.2.1 From VirusTotal queries to AVClass 2

As detailed in Subsection 3.2, we use the SHA256 of the samples in the original EMBER data set to query VirusTotal and get detections from the AV vendors available on VT. We then run AVClass on the collected VirusTotal data. Our primary goal is to get standardized detection names from vendors. Secondly, we obtain a ranking of the generated tags and can categorize them into malware classes (CLASS), families (FAM) or behaviors (BEH).

### 4.2.2 Tag enrichment via co-occurrence

The main output AVClass provides, for each sample: the SHA256 of the file, the number of detections from vendors or a NULL value in the absence of detects, and, finally, a comma-separated list of tags (i.e. of the tag kinds: FILE, BEH, FAM, CLASS, UNK). A score is also attached to each of these tags, as displayed in Listing 1.

```
43211f5628568ae9e25a6011e496663b584d681303a544bc40114936a10764c6 51
FILE:os:windows|11,CLASS:worm|5,FAM:swisyn|5,FAM:reconyc|2,FILE:packed|2
```

Listing 1: Example of AVClass output for a sample.

Alongside individual results, AVClass can also construct a tag co-occurrence matrix, containing all pairs of tags that appear together in samples, along with co-occurrence statistics. Considering a tag pair $(a, b)$, AVClass provides information about: the total number of occurrences of the tag $a$, the total number of occurrences of the tag $b$, the total number of common co-occurrences, common co-occurrences relative to tag $a$, and common co-occurrences relative to tag $b$. Tag frequency information for a given sample is determined only by the total number of distinct samples it appears in, regardless of how many vendors produced this tag. This common co-occurrence frequency can be regarded as a proxy for the relevance of the tag pair, and hence we can utilize it to augment an existing tag list with new tags which frequently appear together. That is to say, if a sample originally has only a tag $x$, but given the co-occurrence matrix we observe that the tag pair $(x, y)$ frequently appears together,

then we will also add tag $y$ to the sample. This enriching mechanism is particularly important in the context of similarity search for Cybersecurity as it allows finding samples which may share several characteristics (i.e. with respect to binary file structure), although they are tagged under different (e.g. family) names – a common practice across AV vendors. The procedure we employ for enriching the EMBER data set with tag co-occurrence information is described in Algorithm 1

---

**Algorithm 1** Tag enrichment algorithm

---

**Input:** Sample with tag metadata, tag co-occurrence matrix, co-occurrence threshold $T$
**Output:** Extra tag info for the sample
1: prev ← previous single tag from AVClass         ▷ From EMBER, 2018
2: curr ← current tag info from AVClass 2         ▷ Mapping from tag to score
3: res ← {}
4: **if not** prev and **not** curr **then**
5:    **return** {}            ▷ Nothing to do, sample is most likely benign
6: **else if** prev and **not** curr **then**
7:    **for** tag pairs (prev → $x$) with $\mathsf{freq}(x \mid \mathtt{prev}) \geq T$ **do**
8:      res[prev, $x$] ← $\mathsf{freq}(x \mid \mathtt{prev})$
9: **else if not** prev and curr **then**
10:    **for** tag $x \in$ curr of kind $\in \{\mathtt{FAM}, \mathtt{CLASS}\}$ **do**
11:      **for** tag pairs ($x \to y$) with $\mathsf{freq}(y \mid x) \geq T$ **do**
12:        res[$x, y$] ← $\mathsf{freq}(y \mid x)$
13: **else**
14:    apply both cases above
15: **return** res

---

### 4.2.3 Obtaining tag ranking for evaluation

We use the rich tag information from AVClass and the co-occurrence analysis, to generate ground truth metadata that can be utilized to evaluate the BCS method based on leaf predictions described in Subsection 4.1. The focus through Subsection 4.2 is on the tag ranking method used to generate the mentioned ground truth metadata. Considering the fact that benign samples do not have tags, we employ a tag ranking procedure applied to all malware samples from the EMBER data set. This procedure takes advantage of supplementary tags acquired through co-occurrence analysis conducted in earlier stages. For understanding the remainder of this subsection, we should recall that in the raw AVClass output, each tag is accompanied by the AVClass rank score.

Given a sample and its AVClass tag list that contains multiple tags of different kinds (FAM, CLASS, BEH, FILE) with their corresponding rank score, conditioned on a tag kind we can derive a probability distribution

$$P(x \mid kind = K) = \frac{\mathsf{score}(x)}{\sum\limits_{\mathsf{tag}\ y\ \mathsf{of\ kind}\ K} \mathsf{score}(y)} \tag{2}$$

We can regard this probability as a measure of confidence and agreement between multiple vendors. Further, we denote by $\mathsf{freq}(x, y)$ the common co-occurrence frequency of tags $x$ and $y$, and by

$$\mathsf{freq}(x \mid y) = \frac{\mathsf{freq}(x, y)}{\mathsf{freq}(y)} \tag{3}$$

the relative occurrence frequency of tag $x$ relative to the occurrences of tag $y$. At the core of the tag ranking algorithms is the formula in Equation (4) which extends the original AVClass rank scores with co-occurrence information:

$$\mathsf{RankScore}(x) = P(x \mid kind = K) + \sum_{y \to x} P(y \mid kind = K) \cdot \mathsf{freq}(x \mid y) \tag{4}$$

Note that due to the scaling by the frequency, this scoring scheme will rank original tags higher than tags obtained via co-occurrence, which may be desirable given that co-occurrence information can be noisier. Algorithm 2 introduces the tag ranking procedure.

---

**Algorithm 2** Tag ranking algorithm

---

    **Input:** Sample with tag `metadata`, tag kind $K$ to rank by
    **Output:** Tag `ranking` for the sample
1: Compute $P(x \mid kind = K)$ for all tags $x \in$ `metadata`
2: Initialize RankScore$[x]$ from $P(x \mid kind = K)$
3: **for** tag pairs $(x \rightarrow y) \in$ `metadata` **do**
4:     **if** $kind(x)$ is `FAM` and $kind(y)$ is $K$ **then**
5:         RankScore$[y]$ += $P(x \mid kind = K) \cdot$ freq$(y \mid x)$

6: `ranking` $\leftarrow$ sort RankScore in descending order and filter to tags of kind $K$
7: **return** `ranking`

---

## 5   Experiments and evaluation

For convenience, we conduct experiments on an architecture that has 64CPUs and 256GB of RAM. A minimal set of requirements to run the method proposed in this paper, consists in being able to load the data set and model in memory. Concretely, for the EMBER data set and the trained XGBoost model the memory consumption is around 27 GB of RAM. Thus, a machine with 32 GB of RAM should be enough to replicate the results in this paper.

Experimentation, in the context of the present work, implies, first of all, determining the ground truth malware tags, as detailed in Subsection 4.2. Secondly, we fine-tune an XGBoost-powered binary malware classifier until it reaches satisfactory performance – i.e. we obtained a ROC AUC score of 0.9966 on EMBER's test split, a value close to the one (i.e. 0.9991) reported in the paper introducing the 2017 batch of EMBER data. Our fine-tuning process yielded the following set of hyperparameters: $max\_depth = 17$, $eta = 0.15$, $n\_estimators = 2048$, $colsample\_bytree = 1$. Finally, also as part of the experimentation, we utilize the leaf predictions to compute the similarity between the samples in EMBER.

Post-experimentation, we perform two different types of evaluation, leveraging both the malicious/benign labels from EMBER (as discussed in Subsection 5.1) and also the tag information computed via AVClass to evaluate the quality of the similarity search as outlined in Subsection 5.2. An important mention is that our evaluation employs a counterfactual analysis scenario in a real-world like setting: we use a test set including an out of time sample, the test data being collected between 2018-11 and 2018-12, while the indexed database only includes samples up until 2018-10. We believe that such a scenario reflects the ever changing nature of the threat landscape as is the case with the Cybersecurity domain.

### 5.1   Top K Selection & Evaluation Based on Malicious/Benign Labels

We select the top K most similar samples from the whole data set, for each query-sample in the test split. Given that the XGBoost classifier did not see the test samples during the training, no bias should be attached to the query process. Our first evaluation scenario is based on counting the labels (malicious/benign) that a query and its most similar samples share. Table 3 shows the results for this type of evaluation considering the top K hits, with K in [1, 10, 50, 100]. The *Mean* column represents the mean hits count with the same class as the needle (e.g. for best performance, the mean should be close to K and the standard deviation should be close to 0). We report ssdeep [25] results in Table 2. In this case, we are relaxing the evaluation condition and consider a hit if the label of the query matches the label of the results and the samples have non-zero ssdeep similarity. The results indicate poor overall performance, with marginally better scores for malicious samples, most likely due to the fact that such samples contain specific byte patterns. In contrast, our proposed method (Table 3) achieves better results for both malicious and benign queries, showing that leaf similarity is well-equipped to find similar entries and becomes more precise as we decrease the value of K.

Table 2: Label homogeneity (ssdeep [25])

| K | Benign | | Malicious | | All | |
|---|---|---|---|---|---|---|
| | Mean | Std | Mean | Std | Mean | Std |
| 1 | 0.51 | 0.49 | 0.8 | 0.39 | 0.66 | 0.47 |
| 10 | 3.55 | 4.34 | 7.31 | 4.16 | 5.43 | 4.65 |
| 50 | 12.31 | 19.24 | 32.59 | 22.12 | 22.45 | 23.07 |
| 100 | 19.84 | 34.82 | 61 | 45.22 | 40.42 | 45.31 |

Table 3: Label homogeneity (leaf similarity)

| K | Benign | | Malicious | | All | |
|---|---|---|---|---|---|---|
| | Mean | Std | Mean | Std | Mean | Std |
| 1 | 1 | 0 | 1 | 0 | 1 | 0 |
| 10 | 9.68 | 1.13 | 9.80 | 0.97 | 9.74 | 1.06 |
| 50 | 47.10 | 7.34 | 48.40 | 6.17 | 47.75 | 6.81 |
| 100 | 92.80 | 16.01 | 96.28 | 13.22 | 94.53 | 14.79 |

## 5.2 Relevance@K Evaluation

The second evaluation flavor explored requires obtaining ground truth metadata in the form of ranking FAM tags as described in Subsection 4.2. Subsequently, given a query sample, we check whether its tag ranking is consistent with the top-K most similar samples. The results of this evaluation scenario are displayed in Table 4.

The underlying assumption followed here is that metadata should be preserved for similar samples. For assessing the relevance between two tag lists, we use the similarity functions listed below:

- **Exact match (EM)** – outputs 1 if the inputs are exactly the same, otherwise outputs 0. It is the strictest comparison method, serving as a lower bound for that performance.
- **Jaccard index (Intersection over Union, IoU)** – disregards element ordering and focuses solely on the presence of items in the inputs.
- **Normalized edit similarity (NES)** – this function, based on Damerau-Levenshtein distance (DL) for strings [52], allows us to penalize differences in rank.

The evaluation procedure, referred to as relevance@K, works as follows: given a query sample and its tag ranking, as well as the top-K most similar hits (each with their own tag rankings), we use the aforementioned functions to calculate the relevance between the query and each hit. This yields K relevance scores for each sample. We then repeat this process for all N query samples in a subset. It is important to note that if a sample is missing its tag list, we consider it to be benign. Therefore, if both the query and hit are missing their tag lists, the relevance is 100%; if the tag list is present for either the query or hit, the relevance is 0%; otherwise, we apply the similarity functions described above.

For constructing the *query* and *knowledge base* subsets, we consider the following scenarios:

1. **Counterfactual analysis** (query = test, knowledge base = train ∪ test). Leveraging the train-test temporal split (i.e. `timestamp(test)` > `timestamp(train)`), we can simulate a real-world production-like scenario, where we use unseen, but labeled data (i.e. the test set) to query the knowledge base.
2. **Unsupervised labeling** (query = unlabelled, knowledge base = train). Given the collection of unlabelled samples, the goal is to identify similar samples within the knowledge base, with the purpose of assigning a provisional label or tag.

Table 4 shows that both scenarios provide very good results in terms of matching queries and hits according to their tags. From a similarity search perspective this means that in the counterfactual analysis scenario simulating a production environment, our method manages to retrieve more than 95% benign similar samples for a given unseen benign query based on tag information and more than 71% relevance of malicious samples for a given malicious query respectively even for 100 hits. This also allows for high confidence unsupervised labelling as can be seen in the bottom table. We do note that the production scenario is expected to be the hardest of the two given that the queries here come

Table 4: Evaluation results accounting for query-hit metadata relevance as given by ranking the FAM tags. The co-occurrence threshold is set to 0.9. The following percentiles are reported, in order, for: $1\%, 10\%, 50\%, 95\%$. In both cases, the query subset size is 200K, evenly split between the labels.

| | | Unsupervised Labelling | | | |
|---|---|---|---|---|---|
| | | Benign | | Malicious | |
| Relevance | Top-K | Mean (Std) | Percentiles | Mean (Std) | Percentiles |
| EM | 1 | 0.986 (0.120) | 0, 1, 1, 1 | 0.717 (0.451) | 0, 0, 1, 1 |
| | 10 | 0.975 (0.097) | 0.5, 0.9, 1, 1 | 0.681 (0.355) | 0, 0.1, 0.8, 1 |
| | 100 | 0.951 (0.124) | 0.34, 0.85, 1, 1 | 0.612 (0.364) | 0, 0.04, 0.7, 1 |
| IoU | 1 | 0.986 (0.120) | 0, 1, 1, 1 | 0.812 (0.349) | 0, 0, 1, 1 |
| | 10 | 0.975 (0.097) | 0.5, 0.9, 1, 1 | 0.780 (0.295) | 0, 0.3, 0.93, 1 |
| | 100 | 0.951 (0.124) | 0.34, 0.85, 1, 1 | 0.714 (0.321) | 0, 0.15, 0.85, 1 |
| NES | 1 | 0.986 (0.120) | 0, 1, 1, 1 | 0.797 (0.353) | 0, 0, 1, 1 |
| | 10 | 0.975 (0.097) | 0.5, 0.9, 1, 1 | 0.764 (0.295) | 0, 0.3, 0.9, 1 |
| | 100 | 0.951 (0.124) | 0.34, 0.85, 1, 1 | 0.697 (0.319) | 0, 0.14, 0.81, 1 |

| | | Counterfactual Analysis | | | |
|---|---|---|---|---|---|
| | | Benign | | Malicious | |
| Relevance | Top-K | Mean (Std) | Percentiles | Mean (Std) | Percentiles |
| EM | 1 | 0.996 (0.066) | 1, 1, 1, 1 | 0.960 (0.196) | 0, 1, 1, 1 |
| | 10 | 0.967 (0.116) | 0.3, 0.9, 1, 1 | 0.730 (0.325) | 0, 0.1, 0.9, 1 |
| | 100 | 0.933 (0.152) | 0.2, 0.78, 1, 1 | 0.641 (0.364) | 0.01, 0.05, 0.77, 1 |
| IoU | 1 | 0.996 (0.066) | 1, 1, 1, 1 | 0.982 (0.093) | 0.5, 1, 1, 1 |
| | 10 | 0.967 (0.115) | 0.3, 0.9, 1, 1 | 0.834 (0.235) | 0.1, 0.5, 0.95, 1 |
| | 100 | 0.933 (0.152) | 0.2, 0.78, 1, 1 | 0.759 (0.290) | 0.01, 0.25, 0.89, 1 |
| NES | 1 | 0.996 (0.066) | 1, 1, 1, 1 | 0.982 (0.093) | 0.5, 1, 1, 1 |
| | 10 | 0.967 (0.115) | 0.3, 0.9, 1, 1 | 0.827 (0.236) | 0.1, 0.48, 0.95, 1 |
| | 100 | 0.933 (0.152) | 0.2, 0.78, 1, 1 | 0.752 (0.292) | 0.01, 0.24, 0.88, 1 |

Table 5: Evaluation results for Mean Average Precision.

| | Unsupervised Labelling | | Counterfactual Analysis | |
|---|---|---|---|---|
| Top-K | Benign | Malicious | Benign | Malicious |
| 1 | 0.986 | 0.716 | 0.996 | 0.960 |
| 10 | 0.985 | 0.753 | 0.985 | 0.895 |
| 50 | 0.973 | 0.714 | 0.967 | 0.812 |
| 100 | 0.966 | 0.698 | 0.957 | 0.783 |

from an out of time distribution compared to the samples the XGBoost model was trained on. For completeness, Table 5 shows the evaluation results with Mean Average Precision (mAP) using exact match to check if two tag rank lists are equivalent (hence relevant).

## 6 Conclusions

**Assumptions and limitations.** The evaluation scheme described in Section 5 requires satisfying two underlying assumptions. Firstly, there are three factors that dictate the quality of the tag information for a sample: (1) the accuracy of VT vendor detections, (2) the tag normalisation and ranking algorithm used by AVClass and (3) the manner in which the tag co-occurrence information is leveraged. Secondly, we consider the lack of tags for a sample to be an indicator that the sample is benign. Although VT detections can be noisy due to various differences across vendors (see section 3), our strategy aims to mitigate this effect as much as possible.

**Potential negative societal impact and ethical concerns.** The data set we enrich with additional similarity-enhancing metadata contains known malware samples and has already benefited Cybersecurity research for many years. Thus Cybersecurity vendors are well aware and capable to defend against all samples included in this data set.

**Future research directions.** We identified two different ideas that can drive future research in this space, while using our current results as support: model interpretability and refining the baseline proposed in this work. For the latter case, we see value in conducting a comparative study, with multiple tree-based models.

**Final remarks.** In the present work, we introduce EMBERSim, a large-scale data set which enriches EMBER with similarity-derived information, with immediate application for sample labeling at two distinct granularities. To the best of our knowledge, we are the first to study the applicability of pairwise similarity, also denoted in this paper as leaf similarity, both in the context of binary code similarity, as well as with applicability to Cybersecurity in general. The aforementioned method shows promising results, as our evaluation indicates (see Section 5). This allows us to empower malware analysts with a comprehensive set of tags describing samples in a pool of malicious and clean samples they can leverage for similarity search at scale. We hope that the insights presented here will enable further research in a domain that would truly benefit from more results – ML-powered binary code similarity in Cybersecurity.

## Acknowledgments and Disclosure of Funding

We thank `Patrick Crenshaw` for the insightful discussions around pairwise similarity. We also thank `Phil Roth` for all the invaluable input and support provided for understanding the contents of the EMBER data set. We are also grateful to `Alexandru Ghiță` and `Ernest Szocs` for their feedback and support throughout the research process.

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

# A  Appendix

## A.1  EMBER data set

The original EMBER2017 data set (as reported in the paper introducing it) contains 1.1M samples, with the date of appearance of more than $95\%$ of them spanning the year 2017. One year later, the authors have collected a second batch of PE files, with $95\%$ of them appearing in 2018. Our experiments follow the related work and are conducted on the most recent batch of EMBER data (from 2018, with feature version 2, as referenced in the EMBER repository [5]).

Figures 2 and 3 show the sample count and label distribution by date of appearance and subset, respectively. EMBER's authors state that the purpose of the 200K unlabelled samples is to aid further research on unsupervised or semi-supervised learning.

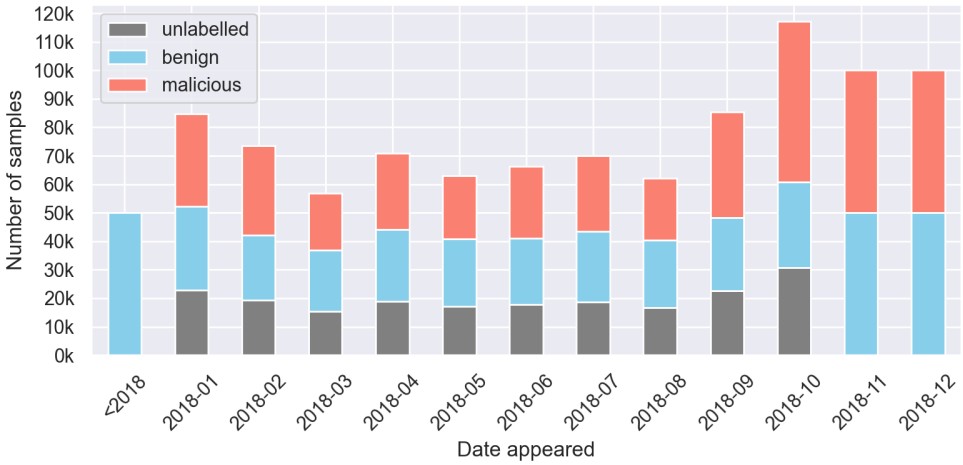

Figure 2: Distribution of sample count per labels with respect to the date of sample first appearance. The vast majority of the samples (95%) appear in 2018.

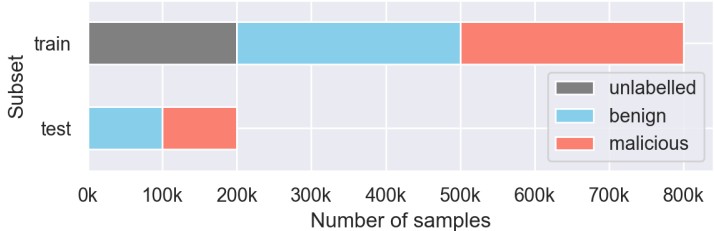

Figure 3: Distribution of sample count per labels with respect to the data subset.

As a measure of confidence in the label, the work introducing EMBER notes that for each malicious sample, more than 40 vendors available in VirusTotal yield a positive detection result. In Figure 4 we show the correlation between the label and the number of VirusTotal detections. The resulting distribution is in line with our expectations, observing a slight fuzziness in the lower regions of detections, nonetheless respecting the 40-vendor threshold empirically set by the authors. We further note that the unlabelled samples appear to be evenly sampled from each detection bin.

---

[5]`https://github.com/elastic/ember`

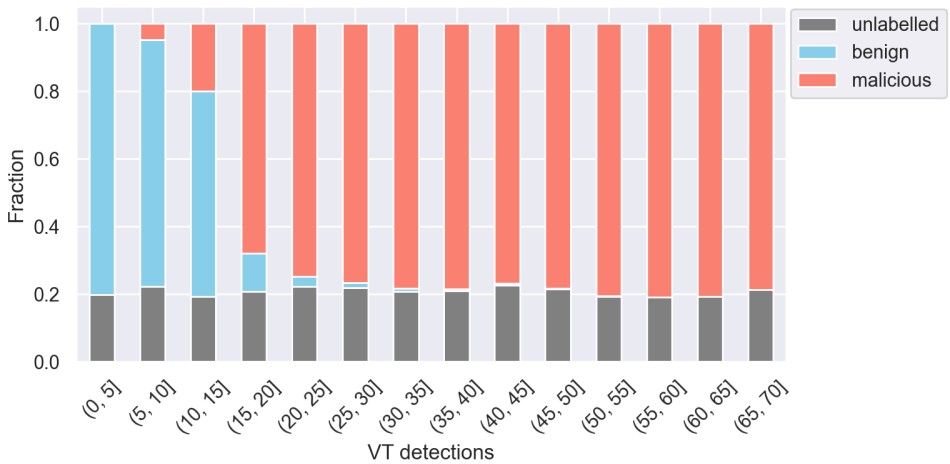

Figure 4: Distribution of VirusTotal detections per label.

## A.2 AVClass2 tags

We turn our attention to metadata obtained from the AVClass2 system [6] for extracting, standardizing and ranking tag information from VirusTotal detections. Particularly of interest for our purposes are the malware family (`FAM`), class (`CLASS`) and behavior (`BEH`) tags, given that they can serve as ground truth for evaluating a similarity search method, under the assumption that the metadata should be consistent between a query and its most similar samples. When applied to all 1M samples in the EMBER2018 data set we explored, AVClass2 was able to identify 1219 unique `FAM` tags, 33 unique `CLASS` tags, and 43 unique `BEH` tags. To offer an intuition related to the order of magnitude of the tag frequency, we show the top 20 most prevalent tags by kind in Figures 5, 6, 7.

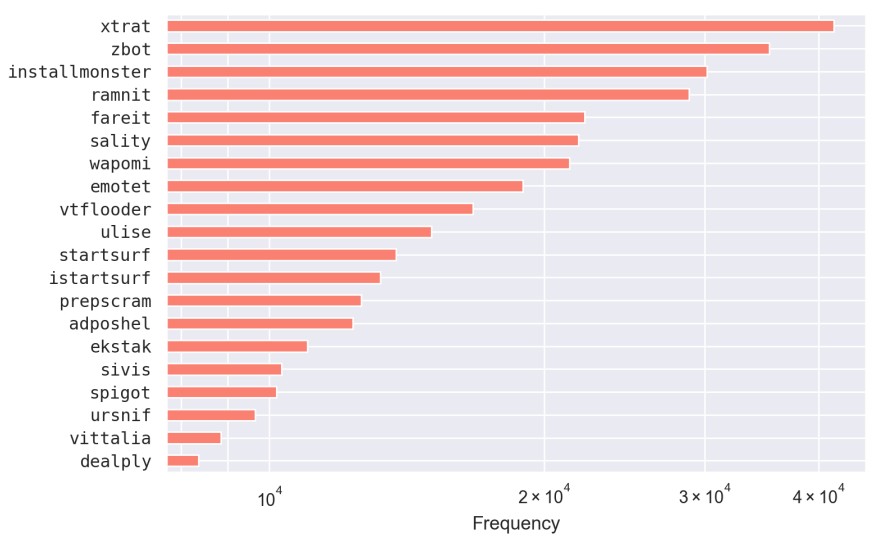

Figure 5: Top 20 most prevalent `FAM` tags (log scale).

---

[6]`https://github.com/malicialab/avclass`

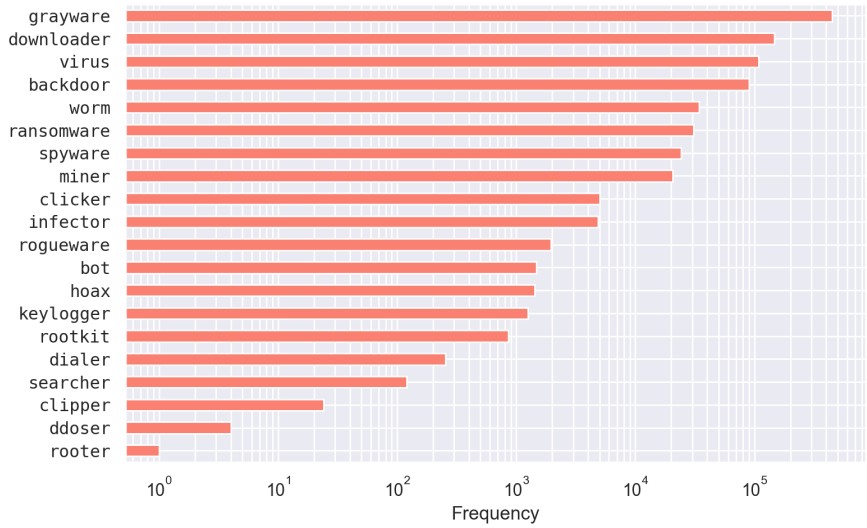

Figure 6: Top 20 most prevalent CLASS tags (log scale).

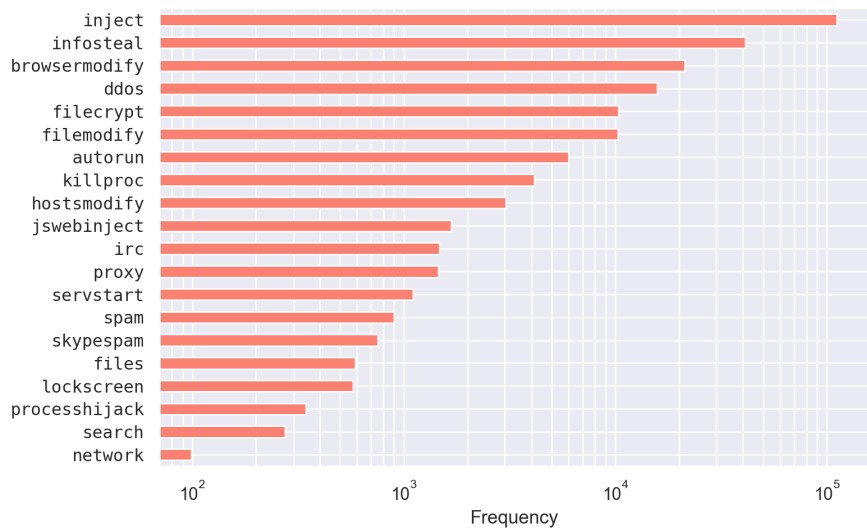

Figure 7: Top 20 most prevalent BEH tags (log scale).

## A.3 Evaluation

### A.3.1 Label homogeneity

Our first evaluation of the similarity search method proposed in this work concerns the ability of preserving the label between a query and its top $K$ hits. We regard this evaluation scenario both as a reasonable assumption for Cybersecurity focused similarity search in general and as a sanity check for the method proposed in this paper. We illustrate the results in Figure 8, with histograms showing the label homogeneity for malicious and benign queries, and their top $K$ hits, with $K \in \{10, 50, 100\}$. We observe that the vast majority of the hits are concentrated in the bin close to $K$, meaning that almost all returned hits share the same label with the query, which is what we would expect for a good performance.

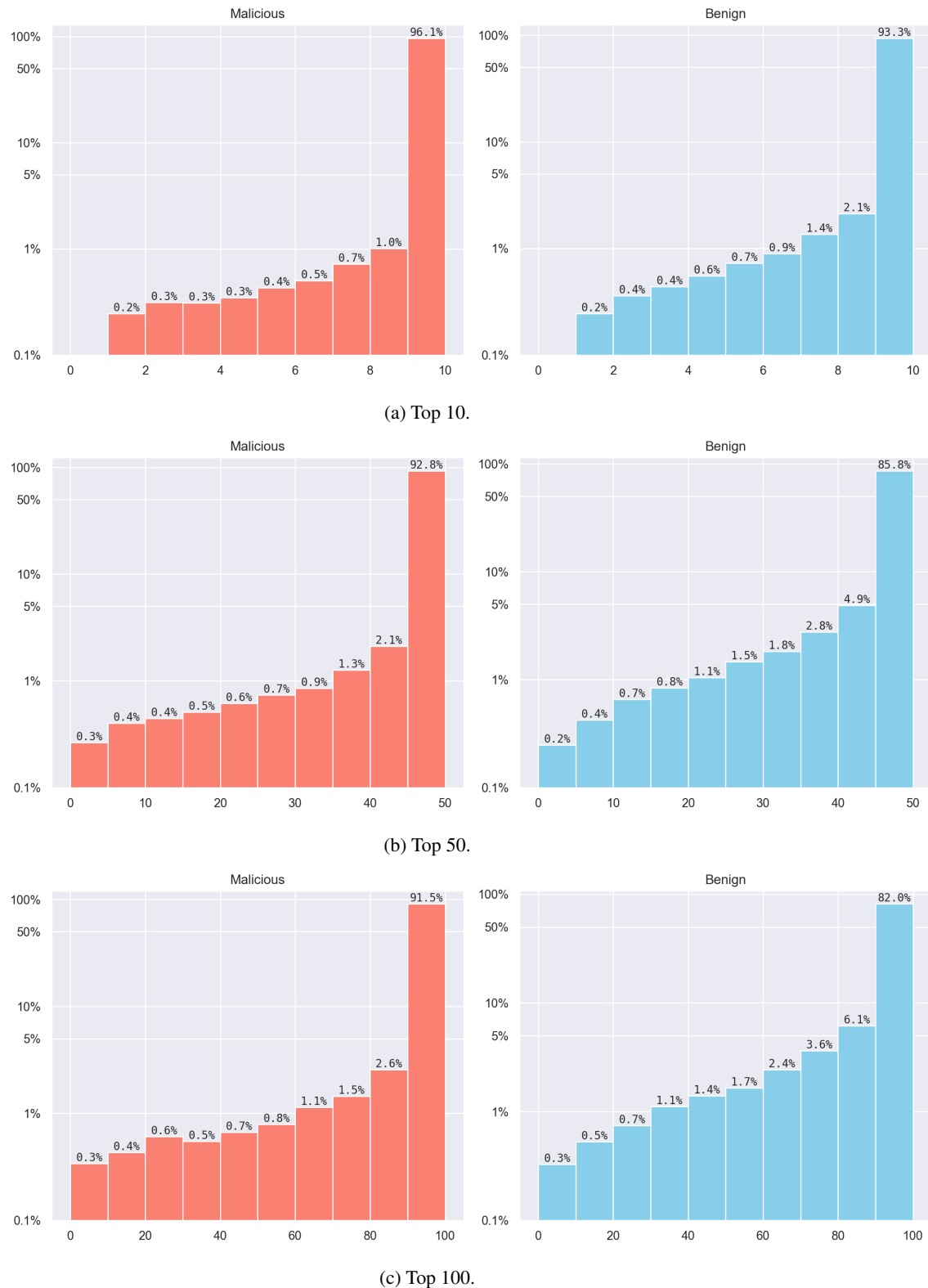

Figure 8: Number of hits (out of top $K$) matching labels with query.

### A.3.2 Relevance @ $K$

Our second evaluation approach implies assessing the relevance between a query and the top $K$ hits as returned by the proposed similarity search method. The relevance @ $K$ objective evaluates the consistency between two metadata lists. Specifically, we check how well the ranking of `FAM` tags for a query is preserved in its top $K$ hits. Results are shown in Figures 9 and 10. We used the following settings: for enriching the tag lists used to construct the tag ranking metadata, we used a common tag co-occurrence threshold of 0.9; in order to penalize inversions in rank, we use the Normalized Edit Similarity function (as described in the main paper). For the histograms, a higher concentration towards $1.0$ is better, meaning that almost all $K$ hits are relevant with respect to the query. For the empirical CDFs, this translates to the curve shifting lower and to the right.

**Counterfactual analysis** (query = test, knowledge base = train $\cup$ test): in which we evaluate the effectiveness of the proposed similarity search method knowing the true label of the query samples.

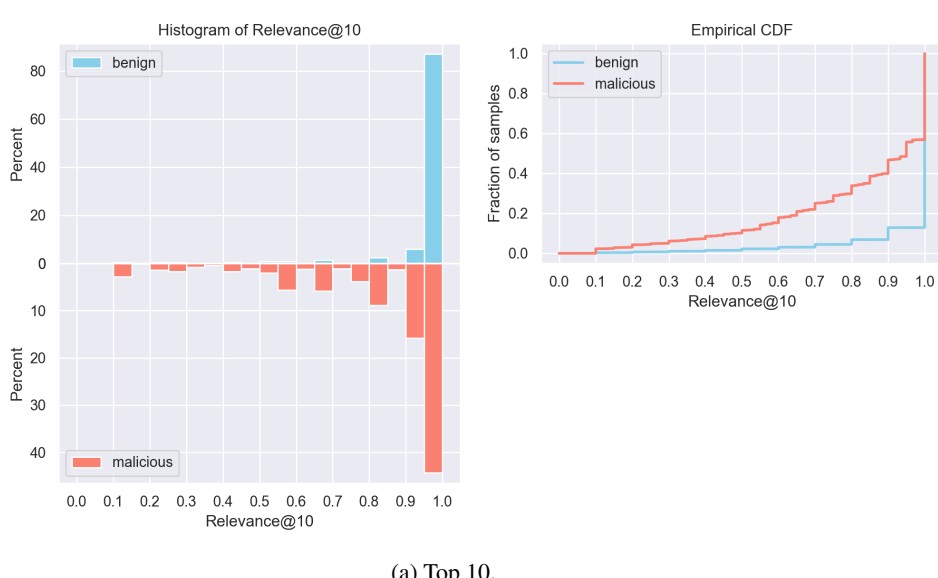

(a) Top 10.

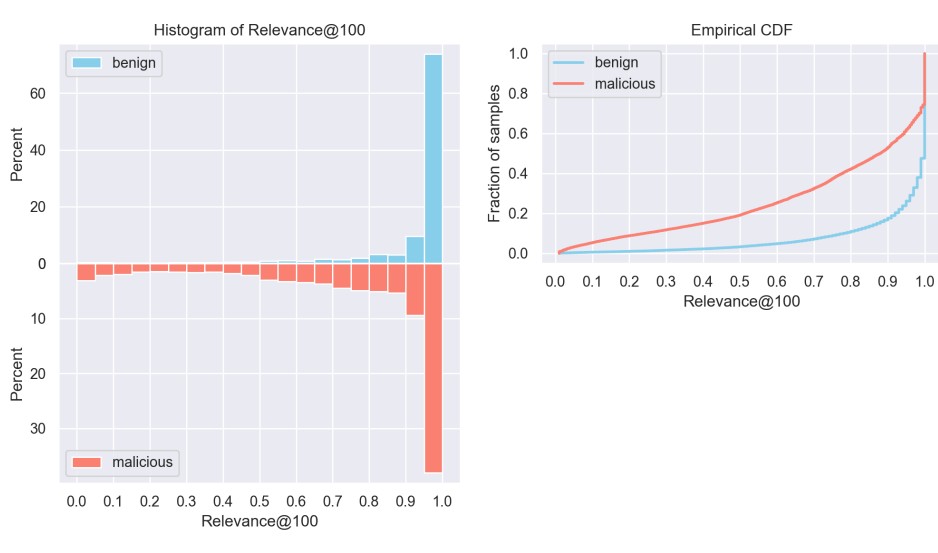

(b) Top 100.

Figure 9: Evaluation results for the *counterfactual analysis* case.

**Unsupervised labelling** (query = unlabelled, knowledge base = train): modelling the scenario where a practitioner wants to label unseen samples using the similarity search method. Since the query set does not contain labels, we leverage the presence of malware tags to serve as an indicator instead.

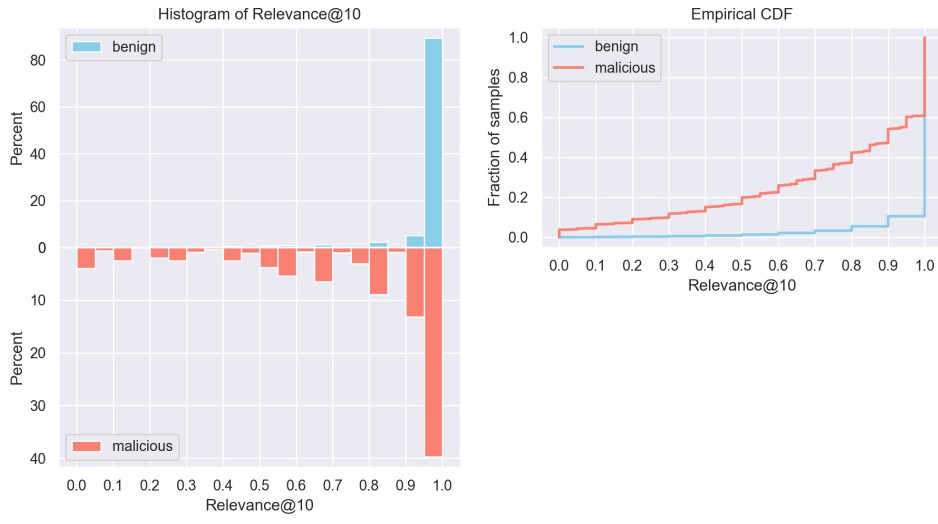

(a) Top 10.

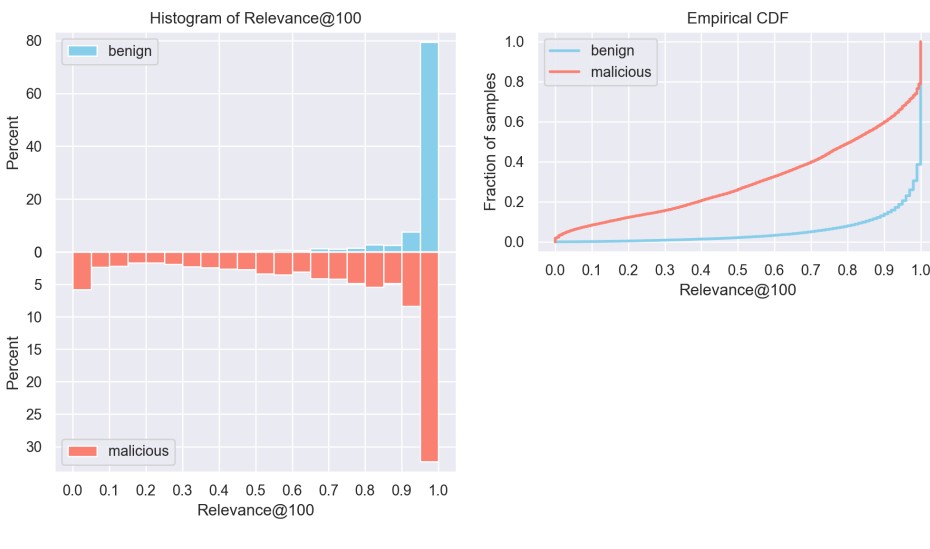

(b) Top 100.

Figure 10: Evaluation results for the *unsupervised labelling* case.

## A.4    Leaf Similarity Extended to other Data Sets

We conduct an additional set of experiments, to address potential generalization concerns that might arise with respect to the leaf similarity approach proposed in this paper. Thus, we use a variant of the DREBIN data set [53]. The DREBIN data set is composed of 129,013 samples, out of which 123,453 are clean and 5,560 malicious.

We perform one-hot encodings on the categorical features in the DREBIN data set, obtaining a total of 550K values. To reduce the amount of computational resources and time required to run this experiment, we apply heuristics (i.e., removing the "*" characters from urls), and then the "hashing trick" [54] to reduce the number of features down to 16,384 values. We perform a 80/20 split of the

data into a train and test subset, while keeping the ratio between clean and dirty samples from the initial DREBIN corpus.

Prior to the final training, we perform hyper-parameter tuning for the XGBoost model, the best results being obtained when the learning rate is 0.05, maximum depth is 12, and number of estimators is equal to 1024. The final XGBoost model obtains a ROC AUC score of 0.95 when evaluated on the test subset. Then we apply the same approach as described in the paper, using leaf similarity class homogeneity check. We obtained the results displayed in Table 6 when considering the "test vs test+train" scenario.

Table 6: Label homogeneity evaluation scenario for leaf similarity over the DREBIN data set.

| K | Benign | | Malicious | | All | |
|---|---|---|---|---|---|---|
| | Mean | Std | Mean | Std | Mean | Std |
| 1 | 1 | 0 | 1 | 0 | 1 | 0 |
| 10 | 9.96 | 0.41 | 8.95 | 2.30 | 9.92 | 0.65 |
| 50 | 49.75 | 2.19 | 38.51 | 16.42 | 49.27 | 4.61 |
| 100 | 99.50 | 4.19 | 71.21 | 35.10 | 98.30 | 10.10 |

As it can be observed from these results, our method achieves good performance on another cybersecurity oriented dataset. This discovery makes us confident that as part of our future work we could enrich our current benchmark based on EMBER with other types of malware data.

### A.5 Discussion on model interpretability

We briefly address model interpretability for our tree-based model, using a TreeSHAP algorithm [55]. We conduct a quick experiment with the end goal of understanding the connection between the features' importance of a group of samples sharing a class tag and the features' importance for the samples that were enriched with the identical tag. We apply the following steps, and compute:

- the SHAP values for the samples with tag X extracted from AVClass;
- the ranking in terms of importance for the features extracted at the previous step;
- the SHAP values as well as the feature importance ranking for the samples with tag X enriched using our method;
- the Spearman correlation between the group feature ranking and the individual sample feature ranking.

Through random sampling, we obtain a moderate correlation, constantly scoring greater than 0.3 which proves that the sample with generated tags are similar from this perspective to the samples that already attached tags from AVClass. Please note that these results were obtained with a tag co-occurrence threshold of 0.9.

We conclude that a line can be drawn between the enriched tags and the feature importance using SHAP values, but the results are not definitive. Additional research needs to be done in the area of interpretability to better understand the model outputs.

Model interpretability represents an interesting future research topic. From this angle, we perform a very narrow assessment of our similarity method, as outlined in this subsection. However, we believe that this represents a separate line of research that should be the primary focus in a future study.

# B Appendix

## B.1 EMBERSim data set documentation

The data used in the databank released as part of this research consists of a postprocessing of the EMBER data set, such that we can easily perform binary code similarity search over this data. The original data set consists of feature vectors from 1 million Windows portable executable (PE) samples. The original EMBER release includes a feature extractor which provides 8 groups of diverse features from each of these PE samples.

In the original formulation, the authors included malicious and benign labels, perfectly balanced among classes along with family tags derived from the first version on the AVClass package. In the most up to date version of the data set only 800K out of 1 million samples are tagged as being either malicious or benign respectively. We use all the conventional features from the original implementation as well as the benign and malicious labels. As part of our contribution, we enhance each sample in EMBER with similarity-informed tags extracted using both an up to date second iteration of the AVClass package (for which we get tags of kinds FAM, CLASS, BEH, FILE) as well as the top 100 most similar samples to a given query according to our leaf similarity method. We provide an extensive evaluation of the similarity method proposed in this paper, showing that it ensures both label and family tag consistency. Thus, this research results in 1 million fully tagged with malware families and labeled (i.e. benign/malicious) samples along with 400k samples for which the top 100 most similar samples are included. Please note that the most similar samples can also be deduced for the rest of the 600K samples based on the information provided. However, our focus is to keep these samples as an indexed database of known samples among which we can search for similar samples.

We propose a split designed to simulate production use cases as closely as possible. We first construct a train split (identical to the EMBER labelled training corpus) on which we propose training similarity informed models. Then we break our queries into two different splits. The first split supports a validation scenario in which we query an indexed database with all the samples seen during training. In this case both the queries and the indexed database span the same time intervals. The second split supports a production-like evaluation scenario, in which we search through an indexed database using queries collected in a time interval disjoint with the interval of the training subset. The splits corresponding to the aforementioned two scenarios, are published using the CSV format.

## B.2 Intended uses

We hope that our proposed contribution will encourage further research in the area of binary code similarity given the large scale corpus we have put together for this purpose. Our intention is to release both our similarity informed tagging technique as well as our similarity method such that upcoming research can construct novel similarity algorithms for binary code using a realistic setting.

## B.3 Data set download

Instructions to download all necessary artifacts as well as reproduce our experiments are provided at: https://github.com/CrowdStrike/embersim-databank. This public GitHub repository links to all the other required open-source resources used in this work and also contains all the code necessary to transform the original data into our format as well as all the sample code required to evaluate our method following the details presented in our paper.

## B.4 Author responsibility for violation of rights

Our proposed contribution does not contain any kind of sensitive data, personally identifiable information or confidential information. The authors are not aware of any possible violation of rights and take responsibility for the published data.

## B.5 Data set hosting and long-term preservation

The authors take full responsibility for the availability of the processed data in the provided repository. No statement can be made about the availability of third-party dependencies such as the EMBER data set or the AVClass repository. However, all the samples (binary files) part of the EMBER data set are

available for download using VirusTotal. VirusTotal is a service free of charge for non-commercial use and is supported by both private companies as well as public contributors as a central source of truth for the cybersecurity community. We make sure to include all the tag information for these samples in our repository which is stored on Zenodo and will continue to be available as long as CERN exists.

### B.6  License

We release our code under a AGPL V3 Clause License, therefore allowing the redistribution and use in source and binary forms, with or without modification as long as the resulting code remains subject to the same license. The metadata is released under a Creative Commons Attribution 4.0 International Public License allowing others to freely share, copy and redistribute the material in any medium or format.

