# OpenReview forum: "EMBERSim: A Large-Scale Databank for Boosting Similarity Search in Malware Analysis"
_NeurIPS.cc/2023/Track/Datasets_and_Benchmarks — NeurIPS 2023 Datasets and Benchmarks Poster_

### Official Review · Reviewer_BNKs · 2023-07-21
**The paper presented a benchmark dataset for binary code similarity by augmenting EMBER. However, the paper did not comprehensively evaluate the quality of the generated tags and benchmark their proposed leaf predictions based similarity against other BCS methods**

**Rating:** 6
**Confidence:** 3
**Correctness:** The reviewer did not observe obvious …
**Clarity:** The presentation of the paper is conc…

**Strengths:**

The paper released an augmented dataset to address the lack of large-scale benchmarking dataset for binary code similarity research. The paper has concise presentation of the proposed procedure for tag augmentation. The paper also proposed two metrics for evaluating the performance of BCS algorithms.

**Additional Feedback:**

No additional feedback

**Documentation:**

The paper has sufficient details on how the tags are augmented. The provided github repository also have detailed information on running the proposed workflows presented in the paper.

**Ethics:**

No.

**Limitations:**

The paper stated the limitation of their tagging process due to missing tags and tagging disagreement between the current and previous versions of AVClass.

**Opportunities For Improvement:**

The adopted a new methodology to quantify binary code similarity using leaf predictions. However, no comparison with other BCS algorithms is attempted in the paper.

**Relation To Prior Work:**

The paper has comprehensive review of related work

**Summary And Contributions:**

The paper published an augmented dataset called EMBERSim by including similarity-derived tags for EMBER for the purpose of benchmarking malware similarity detectors. The paper presented their workflows for tag enrichment in detail.

---

> ### Author Response · Authors · 2023-08-21
> **Response to reviewer BNKs**
>
> We thank the reviewer for the interest in the benchmark, as well as for the insightful comments. We respond to the raised issues below.
>
> Q1: However, no comparison with other BCS algorithms is attempted in the paper.
>
> R1: First of all, a significant part of related works disregard the influence of benign data and even omit it from their similarity datasets [1]. In this work we consider a real-world setting, where defenders must be able to detect similar malware but steer away from producing high amounts of false positives in the process. Thus, we steer away from comparing ourselves with some of the methods referenced above and will instead be looking to present how our method compares against an established line of work in the field based on probabilistic hashing. We present our results below:
> Similar to our counterfactual (out-of-time) evaluation (queries from test and knowledge base train & test), we run ssdeep as a new BCS method on EMBER. When evaluating label homogeneity (checking whether the labels of the top-K results match the label of the query), results show poorer performance of ssdeep compared to our proposed leaf similarity method (e.g. for top-100 we get a mean label match of 19.842 for benign queries and 61 for malicious queries). Furthermore, for benign queries, close to 70% of them have less than 10% non-zero similarity results. This indicates that ssdeep lacks specificity, and is overly sensitive to file modifications, which are more spread for benign samples. For malicious queries, this statistic is slightly better, with approximately 55% percent of results having more than 90% non-zero similarity results. This shows that ssdeep is able to better detect similarities for malicious samples, most likely due to the fact that such samples contain specific byte patterns. Given these findings, we have decided to not further evaluate tag ranking, as we consider it to be a harder problem compared to label homogeneity. Finally, we believe ssdeep to be a poor fit as a BCS method, obtaining inconsistent label results in most similar samples returned.
> Full results:
>
> benign
> | K   | mean  | std   |
> | --  | --    | --    |
> | 1   | 0.51  | 0.49  |
> | 10  | 3.55  | 4.34  |
> | 50  | 12.31 | 19.24 |
> | 100 | 19.84 | 34.82 |
>
> malicious
> | K   | mean  | std   |
> | --  | --    | --    |
> | 1   | 0.80  | 0.39  |
> | 10  | 7.31  | 4.16  |
> | 50  | 32.59 | 22.12 |
> | 100 | 61.00 | 45.22 |
>
> all
> | K   | mean  | std   |
> | --  | --    | --    |
> | 1   | 0.66  | 0.47  |
> | 10  | 5.43  | 4.65  |
> | 50  | 22.45 | 23.07 |
> | 100 | 40.42 | 45.31 |
>
> References:
>
> [1] Brian Ruttenberg, Craig Miles, Lee Kellogg, Vivek Notani, Michael Howard, Charles LeDoux,
> Arun Lakhotia, and Avi Pfeffer. Identifying shared software components to support mal-416
> ware forensics. In Detection of Intrusions and Malware, and Vulnerability Assessment: 11th417
> International Conference, pages 21–40. Springer, 2014

---

### Official Review · Reviewer_6wpR · 2023-07-21
**Extension of malware dataset augmented with similarity score**

**Rating:** 7
**Confidence:** 4
**Clarity:** More clarity/definitions are needed f…

**Strengths:**

The scarcity of malicious data is a main challenge In the cybersecurity domain. The authors have extended an existing dataset with an additional metric, a similarity-based tag, which can be useful in machine learning techniques for finding similar malware.



**Additional Feedback:**

Minor misword, line 253: increasing -> decreasing
Line 128-132: Prevalent values for FILE category is not given
Algorithm 2: Line3-5 Updates the RankScore(y) but do not effect the RankScore(x) which returns at the end.
Formula Number 4 : The summation should show the initial value of the variable.

**Correctness:**

The created dataset is an extension of a popular Dataset. An XGBoost-based method is proposed to identify K similarity and evaluated for different K values. The similarity score for each tag is used for augmenting the original dataset.

**Documentation:**

The code can be found at https://github.com/CrowdStrike/embersim-databank with AGPL-3.0 license. A clear readme is provided. The dataset is available at https://doi.org/10.5281/zenodo.8014709.


**Ethics:**

Did not find any ethic violation

**Limitations:**

No negative impact was identified as the malware is not going to be executed. Although the known malware can be checked for similarity, it would be a good experiment to check for similarity with unknown binary outside the used database.

**Opportunities For Improvement:**

It is worth doing a performance comparison based on other ensamble methods similar to XGBoost such as Random Forrest, GBM etc.
There are standard learning-to-rank models such as RankBoost, RankNet, and LambdaMART.

For ranking performance, metrics such as NDCG, MAP can be used.

**Relation To Prior Work:**

Leaf node prediction is previously exercised in many domains but Cybersecurity and the citations are given in the paper. The proposed work applies the method in Binary code Similarity.

**Summary And Contributions:**

This work presents a large dataset "EMBERSim", an augmented version of "EMBER" dataset by integrating similarity-derived annotations.

---

> ### Author Response · Authors · 2023-08-21
> **Response to reviewer 6wpR**
>
> We kindly thank the reviewer for the detailed feedback on the paper. We respond to the raised issues below.
>
> Q1: It is worth doing a performance comparison based on other ensamble methods similar to XGBoost such as Random Forrest, GBM etc.
> R1: We thank the reviewer for suggesting a performance comparison. As mentioned in the paper, we regard the tree-based similarity as a novel concept in the cybersecurity space and we agree that it is an underdeveloped subject in the area of binary similarity in this space that is ripe for exploration. We feel that for the purposes of a benchmark we should first be striving to compare tree based similarity against more established methods and leave room for researchers to further refine our method if they see potential in it. For this reason, we have also decided to investigate how our leaf similarity method compares to an established binary similarity method based on probabilistic hashing [1] Results are available below:
> Similar to our counterfactual (out-of-time) evaluation (queries from test and knowledge base train & test), we run ssdeep as a new BCS method on EMBER. When evaluating label homogeneity (checking whether the labels of the top-K results match the label of the query), results show poorer performance of ssdeep compared to our proposed leaf similarity method (e.g. for top-100 we get a mean label match of 19.842 for benign queries and 61 for malicious queries). Furthermore, for benign queries, close to 70% of them have less than 10% non-zero similarity results. This indicates that ssdeep lacks specificity, and is overly sensitive to file modifications, which are more spread for benign samples. For malicious queries, this statistic is slightly better, with approximately 55% percent of results having more than 90% non-zero similarity results. This shows that ssdeep is able to better detect similarities for malicious samples, most likely due to the fact that such samples contain specific byte patterns. Given these findings, we have decided to not further evaluate tag ranking, as we consider it to be a harder problem compared to label homogeneity. Finally, we believe ssdeep to be a poor fit as a BCS method, obtaining inconsistent label results in most similar samples returned.
> Full results:
> clean
> | K   | mean  | std   |
> | --  | --    | --    |
> | 1   | 0.51  | 0.49  |
> | 10  | 3.55  | 4.34  |
> | 50  | 12.31 | 19.24 |
> | 100 | 19.84 | 34.82 |
> dirty
> | K   | mean  | std   |
> | --  | --    | --    |
> | 1   | 0.80  | 0.39  |
> | 10  | 7.31  | 4.16  |
> | 50  | 32.59 | 22.12 |
> | 100 | 61.00 | 45.22 |
> all
> | K   | mean  | std   |
> | --  | --    | --    |
> | 1   | 0.66  | 0.47  |
> | 10  | 5.43  | 4.65  |
> | 50  | 22.45 | 23.07 |
> | 100 | 40.42 | 45.31 |
>
> As highlighted above, tree based similarity appears to be competitive against established methods and for this reason we consider your proposals very interesting as a future research direction.
>
> References:
> [1]  Kornblum, Jesse. Identifying almost identical files using context triggered piecewise hashing. Digital investigation, 3, pp.91-97, 2006
>
> Q2: For ranking performance, metrics such as NDCG, MAP can be used.
> R2: Relate EM with MAP for top-1 they are exactly the same, for top10 top100 not sure
> Here are mAP results for:
> Counterfactual analysis (test vs train + test)
> label=0
> | K | mAP |
> | -- | -- |
> | 1 | 0.99567 |
> | 5 | 0.99214 |
> | 10 | 0.98551 |
> | 50 | 0.96734 |
> | 100 | 0.95787 |
>
> label=1
> | K | mAP |
> | -- | -- |
> | 1 | 0.95994 |
> | 5 | 0.93392 |
> | 10 | 0.8957 |
> | 50 | 0.81251 |
> | 100 | 0.78389 |
>
> all
> | K | mAP |
> | -- | -- |
> | 1 | 0.97781 |
> | 5 | 0.96303 |
> | 10 | 0.94061 |
> | 50 | 0.88993 |
> | 100 | 0.87088 |
>
> Unsupervised labeling (unlabeled vs train)
> label=0
> | K | mAP |
> | -- | -- |
> | 1 | 0.98551 |
> | 5 | 0.98837 |
> | 10 | 0.98501 |
> | 50 | 0.97333 |
> | 100 | 0.96671 |
>
> label=1
> | K | mAP |
> | -- | -- |
> | 1 | 0.71666 |
> | 5 | 0.767 |
> | 10 | 0.75355 |
> | 50 | 0.71406 |
> | 100 | 0.69837 |
>
> all
> | K | mAP |
> | -- | -- |
> | 1 | 0.83759 |
> | 5 | 0.86658 |
> | 10 | 0.85767 |
> | 50 | 0.83068 |
> | 100 | 0.81908 |

---

> > ### Author Response · Authors · 2023-08-21
> > **Continuation of response to reviewer 6wpR**
> >
> > Q3:  It would be a good experiment to check for similarity with unknown binary outside the used database.
> > R3: With regards to checking the similarity of unknown binary outside the database we kindly redirect the reviewer to our existing counterfactual analysis. Here, the queries were not present as part of the database used to train our similarity method given that they belong to the original EMBER test set. As can be seen in Fig 1. of our supplementary material, this test set was collected starting from 2018-11 up until 2018-12 after the end date used for collecting the training set of 2018-10. For this reason, we consider the queries to be unknown at training time. We also include the test set as part of our database used for evaluation given that we wish to be able to check for similarity in a realistic scenario. Namely, as new samples come in, they might be similar to the already labeled examples we have seen during training our similarity method, but at the same time, they might only be similar to new unknown samples (which we have not labeled). Given that the original EMBER dataset also includes labels for these out-of-time test samples, it is easy to validate this hypothesis using multiple performance metrics even though these labels as well as these new samples are unknown to the similarity algorithm, given it has not been trained on them.
> >
> > Q4: More clarity/definitions are needed for the variables used in the equations.
> > R4: We thank the reviewer for highlighting minor issues in our equations, we will provide modifications in the revised version of the paper to address these suggestions.

---

> > > ### Comment · Reviewer_6wpR · 2023-08-30
> > >
> > > Thank you for considering my review and your extra effort to provide valuable info. The paper looks strong overall to me and my stance will remain the same.

---

### Official Review · Reviewer_qd4B · 2023-07-22
**Well-done dataset paper, potential hesitation due to usefulness of class-based similarity**

**Rating:** 7
**Confidence:** 3

**Strengths:**

- I looked through their artifact, and it is well and professionally done, including notebooks which demonstrate how the evaluation was performed. The code seems sensible and above the bar (especially for research-level code).

- The topic, binary similarity, is topical and very relevant. EMBER is
  a state-of-the-art dataset in this space. It is very interesting to
  extend it with information that relates to binary similarity, and a
  publicly available dataset of this form could accelerate
  state-of-the-art workloads in this area.

- The paper makes a lot of interesting points and the evaluation is
  done in a thoughtfully-narrative way that was able to be understood
  by just reading the paper in a linear way. The evaluations also
  report reasonable results, as far as I can tell.

- The authors have been thoughtful to account for the fact that EMBER
  has evolved over time, e.g., by using AVClass v2 tags--which were
  not available when EMBER was released.

**Additional Feedback:**

I appreciate the authors thoughtful contribution, and especially thank them for putting together what is a nice artifact that will be useful. I certainly agree that the binary similarity space is ripe for exploration, and undercapitalized. I think the work represents an interesting direction and hope the authors will continue to study more semantically-based methods beyond AVClass.

**Clarity:**

- I found the paper quite easy to read, and I feel that the narrative structure of the paper is fairly nice. Even though I am not an expert in this area, I found it very easy to understand the technical details the authors discuss.

**Correctness:**

- Overall, the dataset looks to be constructed in a sound way. I think I have hit the major points in the other points of my review. The authors' code is available and appears well-structured.

**Documentation:**

Documentation is available

**Ethics:**

There is no ethical concern whatsoever.

**Limitations:**

- The authors have largely thoughtfully addressed limitations. I think they are being fairly transparent with what they are doing early on and clearly identifying that as "binary similarity." I would argue that this is a narrow-minded view, but that may be a matter of taste. I think overall at least the authors use this term consistently, and their limitations section is clear about the major limitations of the work.

- I am not sold on whether class-based similarity is truly useful with respect to other methods, e.g., the heuristic or semantic methods that the authors mention. The authors do not measure against existing ground truth in the area, which to me would seem to be a limitation--perhaps it would be especially limiting if the authors were proposing this as a SOTA innovation in this space, rather than merely a dataset for other researchers to use.

- Following on last point: I think the paper does little to abate the concern around its perspective within the context of the state of the art. There is a lot of implicit assumption that you are buying into this way of analyzing applications. At least for folks who do these kinds of signature and tag-based analyses, I think this is a very reasonable assumption--but it is not universal.

**Opportunities For Improvement:**

- I think there is an implicit question here as to the relevance of
  how the methods proposed will fare compared to other binary
  similarity metrics due to some limitations which the authors have
  nicely addressed. Namely: to care about the work here, I need to
  care about access to tags as assessed by AVClass and similar
  tools. I think this fundamentally limits the scope of "binary
  similarity" to "binary similarity based upon tags from these
  third-party tools." For example, one might think that similarity
  might be assessed based upon something semantically-relevant to the
  binary, e.g., the individual bytes, a higher-level control-flow
  graph, or even some encoding of the bytes as in MalConv. More
  broadly, while the paper calls itself "binary code" similarity,
  there feels a bit of a bait-and-switch, as the binary code is not
  really being analyzed as much as it is having the tags assigned to
  it, correct?

- However, I understand that the above criticism is one of scope, and
  that the authors are instead supposing an audience of researchers
  interested in binary-similarity based upon tags. I think the authors
  are thoughtful in addressing these concerns in the limitations.


**Relation To Prior Work:**

- I think the paper discusses state of the art work in this space well, and it uses modern methods to revive EMBER.

**Summary And Contributions:**

The paper (and associated artifact) provides a new corpus of data
related to binary similarity, building on the popular EMBER dataset.

- The similarity method is based on leaf predictions of tags from AV
  classifiers. The fundamental supposition is that, when construed as
  decision trees, the forest of decisions made by the classifier is
  the same--then the samples are likely to be similar. Additionally,
  there is a notion of closeness, defined by closeness in the vector
  space of associated foci into these trees.

- The authors rank tag closeness in a way that makes sense to me, and
  seems sensible.

- The authors offer their dataset, EMBERSim--an augmentation of EMBER--and it is available.

Overall I mark this paper as accept, though I am on the weaker end due to the limitations that I suggest below. I think the paper is done very thoughtfully and nicely, and the work is publicly available in a nicely-packaged way. I personally think that there is a limited scope of tag-based binary similarity classification, but that does not mean that this is not a valuable contribution, particularly as a dataset paper which can be used by subsequent researchers to use.

---

> ### Author Response · Authors · 2023-08-21
> **Response to reviewer qd4B**
>
> We kindly thank the reviewer for their thorough analysis and observations. We respond to their questions below.
>
> Q1: There is a limited scope of tag-based binary similarity classification
>
> R1: In the related work section we have tried to outline the importance of what we term to be binary code similarity in the cybersecurity context. Given the ever evolving threat landscape, practitioners tend to look for a taxonomy of malware which is constructed via classes and families they consider as tags. The reason for this is that beyond static analysis of a file based on its contents or a dynamic analysis of a sample based on its execution flow, malware belonging to the same family exhibits a similar behavior (even if this behavior is achieved through slightly different means). It is not uncommon for attackers to try and conceal their intentions by modifying a sample such that it will follow a very different and convoluted execution path or will contain a lot of clean-looking code. Having this common language allows researchers to refer to the same entity even if the underlying entity has been tampered with significantly by an attacker.
>
> The VirusTotal platform represents a public forum, where third-party vendors attach tags, generated through leveraging potentially different (even complementary) methods. A variety of  features might be considered in the process, some of which could be very similar to EMBER’s e.g. categorical features, and/or to what the reviewer mentions as being useful for a similarity assessment. We assess similarity based on these tags, while using the tag-enrichment scheme to exercise label agreement between multiple annotators by reconciling their tags into a coherent entity. The method proposed in our work is feature-agnostic and only requires access to samples that can be retrieved from VirusTotal. Thus, with this prerequisite (i.e. VT access), future works can train other binary similarity methods, while having access to a public benchmark of unprecedented scale.

---

> > ### Author Response · Authors · 2023-08-21
> > **Continuation of response to reviewer qd4B**
> >
> > Q2: I am not sold on whether class-based similarity is truly useful with respect to other methods, e.g., the heuristic or semantic methods that the authors mention. The authors do not measure against existing ground truth in the area
> > R2: We agree with the concerns raised and we would like to provide some clarifications. First of all, a significant part of related works disregard the influence of benign data and even omit it from their similarity datasets [1]. In this work we consider a real-world setting, where defenders must be able to detect similar malware but steer away from producing high amounts of false positives in the process. Thus, we steer away from comparing ourselves with some of the methods referenced above and will instead be looking to present how our method compares against an established line of work in the field based on probabilistic hashing. We present our results below:
> >
> > Q3: Following on last point: I think the paper does little to abate the concern around its perspective within the context of the state of the art.
> > R3: As we previously mentioned, we consider that the state of the art in this space suffers from three main limitations:
> > The limited inclusion of clean samples as part of testing in order to be able to gauge the amount of false positives that a given method would generate in a realistic setting
> > The lack of an out of time test, to address the ever evolving threat landscape
> > The limited scale of previous similarity datasets, for comparison (with less than 10K samples in the test set [2,3,1], almost two orders of magnitude below our benchmark)
> > We aim to alleviate the limitations listed above through the inclusion of a large quantity of clean samples (almost 500K) as well as an overall large scale similarity dataset, orders of magnitude larger than previous benchmarks as well as an out of time testing scenario in order to provide a very realistic setting for testing. Our method attains superior results when compared against an established similarity method widely used in the field (i.e. ssdeep). We envision this method as a competitive baseline that future research efforts can use to benchmark their algorithms against. We consider that our reviewers’ proposals could constitute very interesting future research directions which have the potential to nicely benefit the field.
> >
> > References:
> >
> > [1] Brian Ruttenberg, Craig Miles, Lee Kellogg, Vivek Notani, Michael Howard, Charles LeDoux,
> > Arun Lakhotia, and Avi Pfeffer. Identifying shared software components to support mal-416
> > ware forensics. In Detection of Intrusions and Malware, and Vulnerability Assessment: 11th417
> > International Conference, pages 21–40. Springer, 2014
> >
> > [2] Christopher Kruegel, Engin Kirda, Darren Mutz, William Robertson, and Giovanni Vigna. Polymorphic worm detection using structural information of executables. In Recent Advances in Intrusion Detection: 8th International Symposium, pages 207–226. Springer, 2006
> >
> > [3] Saed Alrabaee, Paria Shirani, Lingyu Wang, and Mourad Debbabi. Fossil: a resilient and efficient system for identifying foss functions in malware binaries. ACM Transactions on Privacy and Security, 21(2):1–34, 2018

---

> > > ### Comment · Reviewer_qd4B · 2023-08-24
> > >
> > > I acknowledge the comments. My review remains the same, overall I still see this paper as competitive and I can see that the authors have put in lots of dutiful efforts in really being specific about the details here. I especially appreciate that the authors have done more work in comparing with DREBIN.
> > >
> > > My stance remains the same, I think the paper is well done overall and think the authors have shown commitment to produce a strong final paper.

---

### Official Review · Reviewer_ts2F · 2023-07-25
**EMBERSim dataset review**

**Rating:** 5
**Confidence:** 4
**Clarity:** Yes

**Strengths:**

1. A novel tree-based method for binary code similarity search, which has the potential to significantly improve malware detection.

2. The proposed tag enrichment workflow is highly effective and improves the accuracy of the similarity search.

3. Discuss the proposed method's limitations and potential future research directions. This discussion highlights the need for further research in areas such as improving the accuracy of the similarity search and addressing the challenges of dealing with obfuscated code.

4. The authors provide a clear and detailed description of their experimental setup, including the dataset introduced, the evaluation metrics, and the performance of their method on various benchmarks.


**Additional Feedback:**

I kindly suggest the authors to work on the limitations described above regarding the lack of interpretability or clarity if the proposed workflow can benefit other malware datasets.

**Correctness:**

Yes, the paper well describes the proposed method and its contributions, as well as the limitations and potential applications of the method.

**Documentation:**

Yes, the paper provides a clear description of how data composition processes and experimental setup.

**Ethics:**

No, the EMBER dataset is widely recognized in academic circles, and the malware samples contained within it are generally not considered to be harmful.

**Limitations:**


1. The paper primarily evaluates the proposed method on a single dataset (EMBER), which may not fully represent the diversity and complexity of real-world malware. This limits the generalizability of the authors' findings and raises questions about the effectiveness of the proposed method in real-world settings.

2. Lack of interpretability. While the proposed method achieves high accuracy in detecting similar binary code, it is unclear how the tags generated by the tag enrichment workflow correspond to specific code features or how they would change when changing the XGBoost's hyperparameters.

**Opportunities For Improvement:**

1. The proposed method is evaluated using the EMBER dataset, and it is unclear if it can scale to another malware dataset, e.g., DREBIN.

2. The paper provides a comprehensive review of related work in binary code similarity, which helps contextualize the authors' contributions and highlights the novelty of their approach. However, the authors should emphasize the relevance of malware similarity tasks.

3. The authors could provide more information on the computational resources required to run their method, such as the amount of memory and processing power needed for their tasks.

4. The authors could include more information on the interpretability of their method, such as how the tags generated by the tag enrichment workflow can be used to identify specific features of the binary code that contribute to its similarity with other samples.

5.  Finally, the authors should explore alternative feature extraction and classification methods. What makes XGBoost the best choice? How tuning its parameters would affect the quality of the paper?

**Relation To Prior Work:**

Yes, the paper provides a comprehensive review of related work in the field of malware detection and similarity analysis.

**Summary And Contributions:**

The paper presents EMBERSim, a large-scale data set for boosting similarity search in malware analysis. The authors propose a tree-based method for similarity and a tag enrichment workflow, which allows for high confidence in unsupervised labeling and counterfactual analysis. The contributions of this paper include the creation of EMBERSim, the evaluation of the proposed method, and the demonstration of its potential applications in the field of cybersecurity.

---

> ### Author Response · Authors · 2023-08-21
> **Response to reviewer ts2F**
>
> We kindly thank the reviewer for the detailed feedback, which has helped us improve the presentation of our work. We provide detailed responses to the reviewers' concerns below.
>
> Q1:  It is unclear if it can scale to another malware dataset, e.g., DREBIN.
> R1: Following the reviewer’s suggestion, we use the experimental setting and obtain the results outlined below:
> The DREBIN data set is composed of 129,013 samples, out of which 123,453 are clean and 5,560 malicious. We perform one-hot encodings on the categorical features in the DREBIN data set, obtaining a total of 550K values. To reduce the amount of computational resources and time required to run this experiment, we apply heuristics, and then the “hashing trick” to reduce the number of features down to 16,384 values. We perform a 80/20 split of the data into a train and test subset, while keeping the ratio between clean and dirty samples from the initial DREBIN corpus. Prior to the final training, we perform hyper-parameter tuning for the XGBoost model. Then we apply the same approach as described in the paper, using leaf similarity class homogeneity check. We obtained the following results when considering the “test vs test+train” scenario:
> All
> | K   | Mean   | Std    |
> |-----|--------|--------|
> | 10  | 9.916  | 0.651  |
> | 50  | 49.267 | 4.607  |
> | 100 | 98.298 | 10.097 |
>
>
>
> Class 0
> | K   | Mean   | Std   |
> |-----|--------|-------|
> | 10  | 9.959  | 0.405 |
> | 50  | 49.746 | 2.190 |
> | 100 | 99.503 | 4.191 |
>
>
>
> Class 1
> | K   | Mean   | Std    |
> |-----|--------|--------|
> | 10  | 8.948  | 2.298  |
> | 50  | 38.512 | 16.416 |
> | 100 | 71.214 | 35.097 |
>
>
>
>
> As it can be observed from these results, our method achieves good performance on another cybersecurity oriented dataset. This discovery makes us confident that as part of our future work we could enrich our current benchmark based on EMBER with other types of malware data.
>
> Q2:  The authors should emphasize the relevance of malware similarity tasks.
> R2: We agree with the reviewer that malware similarity tasks are relevant and have many applications in the cybersecurity domain. We insist on this subject in the introductory section of the paper. In this section we emphasize the ease with which attackers can replicate existing malware, hence distributing multiple similar versions of the same content. EMBERSim provides resources that can be used for further research in the area of binary code similarity. As mentioned in the paper, Threat Researchers are in a position in which they have to analyze increasing numbers of similar malicious samples. Thus, Threat Researchers can benefit from the advancements in the area of binary code similarity. Data Scientists training
> ML models for malware detection can perform a more informed sample selection using similarity as a tool towards this. As mentioned in our related work section, the research community regards with increasing interest the subject of malware similarity. Aiming to continue this trend, we release a binary code similarity dataset of unprecedented scale. It is our belief that this resource will provide researchers with enhanced capabilities to tackle binary code similarity.
>
> Q3: The authors could provide more information on the computational resources required to run their method, such as the amount of memory and processing power needed for their tasks.
> R3: For convenience, we ran our experiments on an “g4dn.16xlarge” AWS instance that has 64CPUs and 256GB of RAM. Realistically, any machine that is able to load the dataset and model in memory should be sufficient. Concretely, for the Ember dataset and the trained XGBoost model the memory consumption is around 27 GB of RAM. Thus, a machine with 32 GB of RAM should be enough to replicate the results in the paper.

---

> > ### Author Response · Authors · 2023-08-21
> > **Continuation of response to reviewer ts2F**
> >
> > Q4: What makes XGBoost the best choice?
> >
> > R4: The choice of a tree-based method was driven by the successful applications of leaf predictions to similarity tasks in domains such as Medicine [1, 2] or Biology [3]. We decided to use an XGBoost ensemble due to its relevance in both industry in general [4] and in academic research [5]. While we do not argue that XGBoost is the best approach in this case, we believe that it represents a competitive baseline for this task. Moreover, as highlighted in other comments, the XGBoost powered leaf similarity baseline proposed in this work appears to outperform another popular choice in the field in the form of probabilistic hashing using ssdeep [6]. We agree with the reviewer that varying the parameters of the XGBoost model has the potential to reveal some of the underlying mechanics behind the competitive results we obtain for this method. For the experiments described in the paper, we tune the XGBoost model for binary classification. However we acknowledge that using a weaker model for this task and applying it for computing similarity could represent an interesting future direction.
> >
> > Q5: The paper primarily evaluates the proposed method on a single dataset (EMBER), which may not fully represent the diversity and complexity of real-world malware. This limits the generalizability of the authors' findings and raises questions about the effectiveness of the proposed method in real-world settings.
> >
> > R5: As mentioned above, we have evaluated our method on another dataset as well. We would like to emphasize that even though we apply leaf similarity as a premiere in the cybersecurity field, the method itself has been successfully applied to other domains [1,2,3]. Our purpose for using this method is just to create a competitive baseline for our dataset, we are not trying to obtain a new state of the art. Additionally we would like to mention that through our evaluation, we perform a counterfactual analysis scenario in a real-world like setting: we use a test set including an out of time sample, the test data being collected between 2018-11 and 2018-12, while the indexed database only includes samples up until 2018-10. We believe that such a scenario reflects the ever changing nature of the threat landscape as is the case with the cybersecurity domain.
> >
> > Q6: Lack of interpretability
> >
> > R6: We agree with the concerns raised and we would like to provide some clarifications:
> > A challenging ongoing research topic, model interpretability for tree-based models, can be addressed using TreeSHAP algorithms. We conduct experiments with the end goal of understanding the connection between the features’ importance of a group of samples sharing a class tag and the features’ importance for the samples that were enriched with the identical tag. We apply the following steps, and compute:
> >
> > - the SHAP values for the samples with tag X extracted from AVClass;
> > - the ranking in terms of importance for the features extracted at the previous step;
> > - the SHAP values as well as the feature importance ranking for the samples with tag X enriched using our method;
> > - the Spearman correlation between the group feature ranking and the individual sample feature ranking.
> >
> > Through random sampling, we obtain a correlation score constantly greater than 0.3 (moderate correlation) which proves that the sample with generated tags are similar from this perspective to the samples that already attached tags from AVClass. Please note that these results were obtained with a tag co-occurrence threshold of 0.9. We conclude that, in general, there can be drawn a line between the enriched tags and the feature importance using SHAP values, but the results are not definitive. Additional research needs to be done in the area of interpretability to better understand the model outputs.

---

> > > ### Author Response · Authors · 2023-08-21
> > > **References to support the responses provided**
> > >
> > > References:
> > >
> > > [1] Katherine R Gray, Paul Aljabar, Rolf A Heckemann, Alexander Hammers, Daniel Rueckert
> > > and Alzheimer’s Disease Neuroimaging Initiative. Random forest-based similarity measures for multi-modal classification of alzheimer’s disease. NeuroImage, 65:167–175, 2013
> > >
> > > [2] Katherine R Gray, Paul Aljabar, Rolf A Heckemann, Alexander Hammers, and Daniel Rueckert. Random forest-based manifold learning for classification of imaging data in dementia. In Machine Learning in Medical Imaging: Second International Workshop, pages 159–166. Springer, 2011
> > >
> > > [3] Brian Ruttenberg, Craig Miles, Lee Kellogg, Vivek Notani, Michael Howard, Charles LeDoux, Arun Lakhotia, and Avi Pfeffer. Identifying shared software components to support malware forensics. In Detection of Intrusions and Malware, and Vulnerability Assessment: 11th International Conference, pages 21–40. Springer, 2014
> > >
> > > [4] Ravid Shwartz-Ziv and Amitai Armon. Tabular data: Deep learning is not all you need. Information Fusion, 81:84–90, 2022.
> > >
> > > [5] Leo Grinsztajn, Edouard Oyallon, and Gael Varoquaux. Why do tree-based models still outperform deep learning on typical tabular data? In Thirty-sixth Conference on Neural Information Processing Systems Datasets and Benchmarks Track, 2022.
> > >
> > > [6] Kornblum, Jesse. Identifying almost identical files using context triggered piecewise hashing. Digital investigation, 3, pp.91-97, 2006

---

### Official Review · Reviewer_xndR · 2023-08-30
**Well-constructed benchmark and a good paper {The reviewer is Not an expert in AI for Cybersecurity}**

**Rating:** 7
**Confidence:** 2

**Strengths:**

- Addresses an important problem with a novel technique and useful benchmark dataset.
- Comprehensive metadata enrichment facilitates supervised learning and evaluation.
- Promising results on malware similarity search in initial evaluations.
- Clearly written and technically sound overall.

**Additional Feedback:**

This is a good paper and an admirable contribution overall. A few additional suggestions:

- Provide more implementation details - language, libraries, compute used.
- Highlight the most promising future directions.

Again, the reviewer (myself) would like to disclose themselves as non-experts of AI for Cybersecurity.

**Clarity:**

The paper is well-written overall. The technical approach is well-explained. Figures aid understanding of the workflow.

**Correctness:**

The methodology appears technically sound overall. Claims seem appropriately qualified and evaluation methodology is reasonable. The authors take care to use temporal splits to avoid training/test contamination.

**Documentation:**

For the EMBERSim dataset submission, the authors provide a reasonable level of detail on the data collection, organization, and intended uses. The data sources, extraction process, and metadata enrichment workflows are described. Ethical usage of malware data is discussed. The dataset is hosted on Zenodo with a clear license, and maintenance through the authors' institution is implied. For the benchmark, some additional implementation and computational details would be beneficial for reproducibility. The code is open source which aids replication, but more specifics on languages, frameworks, hyperparameter tuning, and hardware used would make the benchmark easier to reproduce. Overall, the documentation is reasonable but expanded details, especially around reproducibility of the benchmark technique, would strengthen it.

**Ethics:**

I do not see any major ethical issues with this work. The data seems appropriate for malware research and the authors state there is no sensitive information present. Potential negative impacts are considered.

**Limitations:**

The authors clearly acknowledge the noisiness of AV vendor detections and the limitations of the tagging approach. They suggest interpretability analysis in future work to better understand the model, which could help address these issues. Overall, the limitations are adequately addressed.

**Opportunities For Improvement:**

- More analysis could be provided on the labeling accuracy of AVClass and impact of tag co-occurrence threshold.
- Additional baselines beyond ssdeep would help better characterize performance.
- More details could be provided in the main text on computational requirements and concrete guidelines for reproducibility (which seems already updated in the revision).

**Relation To Prior Work:**

Relevant literature is covered. The novelty is articulated, although comparisons to other similarity techniques are lacking.

**Summary And Contributions:**

This paper introduces EMBERSim, an enhanced EMBER dataset with similarity information to enable binary code similarity research. The key contributions are a large-scale benchmark, a novel similarity method using XGBoost leaf predictions, and comprehensive metadata enrichment. The work aims to enable more research on binary code similarity for malware analysis, which a lack of suitable datasets has limited. Overall, this is a valuable contribution to the malware research community.

---

> ### Author Response · Authors · 2023-08-30
>
> We thank the reviewer for their time and feedback. We agree with the proposed opportunities for improvement and we particularly acknowledge the need for a more comprehensive study for refining the baselines.
>
> > Q1: Provide more implementation details - language, libraries, compute used.
>
> We kindly refer the reviewer to the open source implementation, where we provide a list of libraries and packages needed to reproduce our experiments. The computational requirements are described in Section 5 of the revised version of the paper.
>
> > Q2: Highlight the most promising future directions.
>
> In the revised version of the paper, we introduced a paragraph acknowledging the most important research directions building upon our work, namely:  widening the comparative study and interpretability (both from the perspective of the underlying model and from the tagging workflows).

---

### Author Response · Authors · 2023-08-21
**General Comment**

First of all, we would like to thank the reviewers for their time and valuable suggestions. We are pleased to see that the reviewers acknowledge the importance of our work in the binary code similarity space with applicability to the cybersecurity domain. Furthermore, the reviewers  understand the value of the resources introduced in this work, with focus on the benchmark and method based on leaf predictions that we used, in premiere, to address binary code similarity.

Although the reviewers appreciate the premise and artifacts resulting from this work, they expressed some concerns regarding the interpretability of our tag enrichment workflow and proposed some interesting new research directions in order to clarify if the proposed workflow could benefit other malware datasets. Furthermore, the reviewers suggested comparing our baseline against other known methods that are used to quantify binary similarity in cybersecurity settings. We are grateful for the reviewers’ numerous questions that have helped us identify improvements in the presentation of our experimental setting and results.

We address each reviewer’s questions and concerns in a separate comment. We will include all the details and clarifications provided as part of our responses in the revised version of the paper we will include on the OpenReview platform. In summary, among the main improvements included in the new version, we count:

- We provide more context around the variables used in some of our equations for clarity
- To emphasize the generalization capabilities of leaf similarity, we evaluate this method on  another data set
- We compare the leaf similarity method proposed in this work, against a popular BCS method that is based on probabilistic hashing. Results show that our method is marginally better
- We provide additional clarifications around our counterfactual analysis scenario and how it touches on a real world setting given our out of time problem formulation

We hope that the revised version of the paper adequately addresses the concerns and implements the suggestions that the reviewers have expressed.

---

### Decision · Program_Chairs · 2023-09-22

**Decision:**

Accept (Poster)

**Comment:**

This paper focuses on the domain of malware analysis, and extends the popular EMBER dataset to enable better research on binary similarity. The authors publish an augmented dataset based on EMBER, which includes similarity-derived tags for benchmarking malware similarity detectors.

Reviewers agree that the paper is well written, clear and thorough, and that the authors do a good job of clearly defining and discussing its scope and limitations.

The artifacts included with the paper are well done, and should have a real positive impact on research in this area.

The recommendation is accept.